# New Fixed Point Theorems with Applications to Non-Linear Neutral Differential Equations

**Laila A. Alnaser [1], Jamshaid Ahmad [2,\*], Durdana Lateef [1] and Hoda A. Fouad [1,3]**

[1]  Department of Mathematics, College of Science, Taibah University,
    Al Madina Al Munawara 41411, Saudi Arabia; alnaser_layla@yahoo.com (L.A.A.);
    drdurdanamaths@gmail.com (D.L.); hoda_rg@yahoo.com (H.A.F.)
[2]  Department of Mathematics, University of Jeddah, P.O. Box 80327, Jeddah 21589, Saudi Arabia
[3]  Faculty of Science, Alexandria University, Alexandria 21500, Egypt
\*  Correspondence: jkhan@uj.edu.sa; Tel.: +966-569765680

**Abstract:** The aim of this study is to investigate the existence of solutions for a non-linear neutral differential equation with an unbounded delay. To achieve our goals, we take advantage of fixed point theorems for self-mappings satisfying a generalized $(\alpha, \varphi)$ rational contraction, as well as cyclic contractions in the context of $\mathcal{F}$-metric spaces. We also supply an example to support the new theorem.

**Keywords:** nonlinear neutral differential equation; $\mathcal{F}$-metric space; $(\alpha, \varphi)$ rational contraction; fixed point

## 1. Introduction

The concept of a metric space was initiated by Frechet [1] in the following way: A metric on a non-empty set $\mathcal{S}$ is a mapping $d : \mathcal{S} \times \mathcal{S} \to [0, +\infty)$ satisfying the following properties:

(i)    $d(\mathfrak{u}, \mathfrak{v}) = 0 \Longleftrightarrow \mathfrak{u} = \mathfrak{v}$,
(ii)   $d(\mathfrak{u}, \mathfrak{v}) = d(\mathfrak{v}, \mathfrak{u})$, and
(iii)  $d(\mathfrak{u}, \mathfrak{w}) \leq d(\mathfrak{u}, \mathfrak{v}) + d(\mathfrak{v}, \mathfrak{w})$,

for all $\mathfrak{u}, \mathfrak{v}, \mathfrak{w} \in \mathcal{S}$. The pair $(\mathcal{S}, d)$ is called a metric space. Many interesting generalizations (or extensions) of the metric space have recently appeared. Czerwik [2], Branciari [3], and Matthews [4] initiated the notions of $b$-metric spaces, generalized metric spaces, and partial metric spaces reseptively. Very recently, Jleli et al. [5] introduced an attractive generalization of a metric space, as follows.

Suppose that $\mathcal{F}$ is a set of functions $f : (0, +\infty) \to \mathbb{R}$ satisfying the assertions:

$(\mathcal{F}_1)$ $f$ is non-decreasing, and
$(\mathcal{F}_2)$ For each sequence $\{\alpha_n\} \subseteq R^+$, $\lim_{n \to \infty} \alpha_n = 0$ if and only if $\lim_{n \to \infty} f(\alpha_n) = -\infty$.

**Example 1.** *The following functions $f : (0, +\infty) \to \mathbb{R}$ are elements of $\mathcal{F}$.*

- $f(\theta) = \ln(\theta)$,
- $f(\theta) = \theta + \ln(\theta)$,
- $f(\theta) = -\frac{1}{\sqrt{\theta}}$, and
- $f(\theta) = \ln(\theta^2 + \theta)$,

for $\theta > 0$.

**Definition 1.** *[5] Let $\mathcal{S} \neq \varnothing$, and let $d_{\mathcal{F}} : \mathcal{S} \times \mathcal{S} \to [0, +\infty)$ be a given function. Suppose that $\exists\, (f, h) \in \mathcal{F} \times [0, +\infty)$, such that*

(D₁) $(\mathfrak{u}, \mathfrak{v}) \in \mathcal{S} \times \mathcal{S}, d_{\mathcal{F}}(\mathfrak{u}, \mathfrak{v}) = 0 \iff \mathfrak{u} = \mathfrak{v}$.
(D₂) $d_{\mathcal{F}}(\mathfrak{u}, \mathfrak{v}) = d_{\mathcal{F}}(\mathfrak{v}, \mathfrak{u})$, for all $(\mathfrak{u}, \mathfrak{v}) \in \mathcal{S} \times \mathcal{S}$.
(D₃) For each $(\mathfrak{u}, \mathfrak{v}) \in \mathcal{S} \times \mathcal{S}$, for each $N \in \mathbb{N}$ ($N \geq 2$), and for every $(\mathfrak{w}_i)_{i=1}^{N} \subset \mathcal{S}$ such that $(\mathfrak{w}_1, \mathfrak{w}_N) = (\mathfrak{u}, \mathfrak{v})$, we have

$$d_{\mathcal{F}}(\mathfrak{u}, \mathfrak{v}) > 0 \Rightarrow f(d_{\mathcal{F}}(\mathfrak{u}, \mathfrak{v})) \leq f\left(\sum_{i=1}^{N-1} d_{\mathcal{F}}(\mathfrak{u}_i, \mathfrak{u}_{i+1})\right) + h. \tag{1}$$

*Then, $d_{\mathcal{F}}$ is called an $\mathcal{F}$-metric on $\mathcal{S}$, and $(\mathcal{S}, d_{\mathcal{F}})$ is called an $\mathcal{F}$-metric space.*

**Remark 1.** *This notion of a $\mathcal{F}$-metric is more comprehensive than the notion of a standard metric, because any metric $d$ is an $\mathcal{F}$-metric $d_{\mathcal{F}}$; however, the converse is not true.*

**Example 2.** *[5] The space $(\mathbb{R}, d_{\mathcal{F}})$ is an $\mathcal{F}$-metric space, with the $\mathcal{F}$-metric $d_{\mathcal{F}}$ defined by*

$$d_{\mathcal{F}}(\mathfrak{u}, \mathfrak{v}) = \begin{cases} (\mathfrak{u} - \mathfrak{v})^2 & \text{if } (\mathfrak{u}, \mathfrak{v}) \in [0, 3] \times [0, 3] \\ |\mathfrak{u} - \mathfrak{v}| & \text{if } (\mathfrak{u}, \mathfrak{v}) \notin [0, 3] \times [0, 3] \end{cases}$$

*with $f(\theta) = \ln(\theta)$ and $h = \ln(3)$.*

**Definition 2.** *[5] Suppose that $(\mathcal{S}, d_{\mathcal{F}})$ is an $\mathcal{F}$-metric space.*

(i) A sequence $\{\mathfrak{u}_n\}$ in $\mathcal{S}$ is said to be $\mathcal{F}$-convergent to some $\mathfrak{u} \in \mathcal{S}$ if $\{\mathfrak{u}_n\}$ is convergent to $\mathfrak{u}$, with respect to $d_{\mathcal{F}}$.
(ii) $\{\mathfrak{u}_n\}$ is said to be $\mathcal{F}$-Cauchy, if

$$\lim_{n,m \to \infty} d_{\mathcal{F}}(\mathfrak{u}_n, \mathfrak{u}_m) = 0.$$

(iii) If each $\mathcal{F}$-Cauchy sequence in $\mathcal{S}$ is $\mathcal{F}$-convergent, then $(\mathcal{S}, d_{\mathcal{F}})$ is $\mathcal{F}$-complete.

**Theorem 1.** *[5] Let $(\mathcal{S}, d_{\mathcal{F}})$ be an $\mathcal{F}$-metric space and $\mathcal{J} : \mathcal{S} \to \mathcal{S}$. Suppose that the following assertions are satisfied:*

(i) $(\mathcal{S}, d_{\mathcal{F}})$ is $\mathcal{F}$-complete, and
(ii) $\exists\, k \in (0, 1)$ such that
$$d_{\mathcal{F}}(\mathcal{J}(\mathfrak{u}), \mathcal{J}(\mathfrak{v})) \leq k d_{\mathcal{F}}(\mathfrak{u}, \mathfrak{v}). \tag{2}$$

*Then $\mathcal{J}$ has a unique fixed point $\mathfrak{u}^* \in \mathcal{S}$. Moreover, for any $\mathfrak{u}_0 \in \mathcal{S}$, the sequence $\{\mathfrak{u}_n\} \subset \mathcal{S}$ defined by*

$$\mathfrak{u}_{n+1} = g(\mathfrak{u}_n), \quad n \in \mathbb{N},$$

*is $\mathcal{F}$-convergent to $\mathfrak{u}^*$.*

Furthermore, Alnaser et al. [6] and Lateef et al. [7] obtained the relation-theoretic contraction results and the fixed point theorems of Dass and Gupta, respectively, owing to the notion of $\mathcal{F}$-metric spaces.

## 2. Materials and Methods

In this paper, we employ an $\mathcal{F}$-metric space $(\mathcal{S}, d_{\mathcal{F}})$ and a self mapping $\mathcal{J} : \mathcal{S} \to \mathcal{S}$ to define some generalized contractions, named $(\alpha, \varphi)$ rational contractions and cyclic contractions. A family of non-decreasing functions $\varphi : [0, +\infty) \to [0, +\infty)$ satisfying $\sum_{n=1}^{\infty} \varphi^n(t) < +\infty$ for all $t > 0$, where $\varphi^n$ is

the *n*-th iterate of $\varphi$ and a real-valued function $\alpha : \mathcal{S} \times \mathcal{S} \to [0, +\infty)$ with the concept of $\alpha$-admissibility is used. To verify the effectiveness and applicability of our main results, the solution of a differential equation is also manipulated.

We define a sequence $\{\mathfrak{u}_n\}$ in $\mathcal{S}$ by $\mathfrak{u}_{n+1} = \mathcal{J}(\mathfrak{u}_n)$ for all $n \in \mathbb{N}$. By using the different assumptions given in the contractive conditions, we establish that $\{\mathfrak{u}_n\}$ is a Cauchy sequence. We take $(\mathcal{S}, d_{\mathcal{F}})$ to be $\mathcal{F}$-complete, and so $\{\mathfrak{u}_n\}$ converges to a point $\mathfrak{u}^* \in \mathcal{S}$. Then, we prove that $\mathfrak{u}^*$ is a fixed point of the mapping $\mathcal{J} : \mathcal{S} \to \mathcal{S}$.

## 3. Results

The aim of this section is to define $(\alpha, \varphi)$ rational contractions and then cyclic contractions in the setting of $\mathcal{F}$-metric spaces, in order to establish some new fixed point results.

### 3.1. Fixed Point Result for $(\alpha, \varphi)$ Rational Contractions

In 2012, Samet et al. [8] initiated the notions of $\alpha$-admissible mappings and $(\alpha, \varphi)$ contractive mappings and proved various fixed point theorems for such mappings.

Consistent with Samet et al. [8], $\Psi$ denotes the family of non-decreasing functions $\varphi : [0, +\infty) \to [0, +\infty)$ such that $\sum_{n=1}^{\infty} \varphi^n(t) < +\infty$ for all $t > 0$, where $\varphi^n$ is the *n*-th iterate of $\varphi$.

**Lemma 1.** *[8] If $\varphi \in \Psi$, then we have the following:*

(i) $(\varphi^n(t))_{n \in \mathbb{N}}$ *converges to 0 as $n \to \infty$ for all $t \in (0, +\infty)$;*
(ii) $\varphi(t) < t$ *for all $t > 0$; and*
(iii) $\varphi(t) = 0$ *iff $t = 0$.*

**Definition 3.** *[8] Let $\mathcal{J} : \mathcal{S} \to \mathcal{S}$ and $\alpha : \mathcal{S} \times \mathcal{S} \to [0, +\infty)$. Then, $\mathcal{J}$ is said to be $\alpha$-admissible if*

$$\mathfrak{u}, \mathfrak{v} \in \mathcal{S}, \quad \alpha(\mathfrak{u}, \mathfrak{v}) \geq 1 \quad \Longrightarrow \quad \alpha(\mathcal{J}\mathfrak{u}, \mathcal{J}\mathfrak{v}) \geq 1.$$

**Theorem 2.** *[8] Let $(\mathcal{S}, d)$ be a complete metric space and $\mathcal{J}$ be an $\alpha$-admissible mapping. Assume that*

$$\alpha(\mathfrak{u}, \mathfrak{v}) d(\mathcal{J}\mathfrak{u}, \mathcal{J}\mathfrak{v}) \leq \varphi(d(\mathfrak{u}, \mathfrak{v}))$$

*for all $\mathfrak{u}, \mathfrak{v} \in \mathcal{S}$, where $\varphi \in \Psi$. Also, suppose that*

(i) *there exists $\mathfrak{u}_0 \in \mathcal{S}$ such that $\alpha(\mathfrak{u}_0, \mathcal{J}\mathfrak{u}_0) \geq 1$; and*
(ii) *either $\mathcal{J}$ is continuous or, for any sequence $\{\mathfrak{u}_n\}$ in $\mathcal{S}$ such that $\alpha(\mathfrak{u}_n, \mathfrak{u}_{n+1}) \geq 1$ for all $n \in \mathbb{N}$ and $\mathfrak{u}_n \to \mathfrak{u}$ as $n \to +\infty$, we have $\alpha(\mathfrak{u}_n, \mathfrak{u}) \geq 1$ for all $n \in \mathbb{N}$.*

*Then, $\mathcal{J}$ has a fixed point.*

For more details on $(\alpha, \varphi)$ contractions, we refer the reader to [12–17].

**Definition 4.** *Let $(\mathcal{S}, d_{\mathcal{F}})$ be an $\mathcal{F}$-metric space. The mapping $\mathcal{J} : \mathcal{S} \to \mathcal{S}$ is said to be an $(\alpha, \varphi)$ rational contraction if there exist two functions $\alpha : \mathcal{S} \times \mathcal{S} \to [0, +\infty)$ and $\varphi \in \Psi$, such that*

$$\alpha(\mathfrak{u}, \mathfrak{v}) d_{\mathcal{F}}(\mathcal{J}(\mathfrak{u}), \mathcal{J}(\mathfrak{v})) \leq \varphi(\mathcal{R}(\mathfrak{u}, \mathfrak{v})), \tag{3}$$

*where*

$$\mathcal{R}(\mathfrak{u}, \mathfrak{v}) = \max\{d_{\mathcal{F}}(\mathfrak{u}, \mathfrak{v}), \frac{d_{\mathcal{F}}(\mathfrak{u}, \mathcal{J}(\mathfrak{u})) d_{\mathcal{F}}(\mathfrak{v}, \mathcal{J}(\mathfrak{v}))}{1 + d_{\mathcal{F}}(\mathfrak{u}, \mathfrak{v})}\} \tag{4}$$

*for $\mathfrak{u}, \mathfrak{v} \in \mathcal{S}$.*

**Theorem 3.** *Let $(\mathcal{S}, d_{\mathcal{F}})$ be an $\mathcal{F}$-metric space and $\mathcal{J} : \mathcal{S} \to \mathcal{S}$ be both an $(\alpha, \varphi)$ rational contraction and $\alpha$-admissible. Suppose that the following assertions are satisfied:*

(i) *$(\mathcal{S}, d_{\mathcal{F}})$ is $\mathcal{F}$-complete,*
(ii) *there exists $u_0 \in \mathcal{S}$ such that $\alpha(u_0, \mathcal{J}(u_0)) \geq 1$, and*
(iii) *if $\{u_n\}$ is a sequence in X, such that $\alpha(u_n, u_{n+1}) \geq 1$ for all $n$ and $u_n \to u^* \in \mathcal{S}$ as $n \to \infty$, then $\alpha(u_n, u^*) \geq 1$ for all $n \in \mathbb{N}$.*

*Then, $\mathcal{J}$ has a fixed point $u^* \in \mathcal{S}$*

**Proof.** Let $u_0 \in \mathcal{S}$ such that $\alpha(u_0, \mathcal{J}(u_0)) \geq 1$. Define a sequence $\{u_n\}$ in $\mathcal{S}$ by $u_{n+1} = \mathcal{J}(u_n)$ for all $n \in \mathbb{N}$. If $u_{n+1} = u_n$ for some $n \in \mathbb{N}$, then $u^* = u_n$ is a fixed point of $\mathcal{J}$. So, we assume that $u_{n+1} \neq u_n$ for all $n \in \mathbb{N}$. Then, as $\mathcal{J}$ is $\alpha$-admissible, we get $\alpha(u_0, u_1) = \alpha(u_0, \mathcal{J}(u_0)) \geq 1$ implies $\alpha(u_1, u_2) = \alpha(\mathcal{J}(u_0), \mathcal{J}(u_1)) \geq 1$. By induction, we get $\alpha(u_n, u_{n+1}) \geq 1$ for all $n \in \mathbb{N}$. By (3) with $u = u_{n-1}$ and $v = u_n$, we have

$$d_{\mathcal{F}}(u_n, u_{n+1}) = d_{\mathcal{F}}(\mathcal{J}(u_{n-1}), \mathcal{J}(u_n)) \leq \alpha(u_{n-1}, u_n) d_{\mathcal{F}}(\mathcal{J}(u_{n-1}), \mathcal{J}(u_n)) \leq \varphi(\mathcal{R}(u_{n-1}, u_n)), \quad (5)$$

where

$$
\begin{aligned}
\mathcal{R}(u_{n-1}, u_n) &= \max\{d_{\mathcal{F}}(u_{n-1}, u_n), \frac{d_{\mathcal{F}}(u_{n-1}, \mathcal{J}(u_{n-1})) d_{\mathcal{F}}(u_n, \mathcal{J}(u_n))}{1 + d_{\mathcal{F}}(u_{n-1}, u_n)}\} \\
&= \max\{d_{\mathcal{F}}(u_{n-1}, u_n), \frac{d_{\mathcal{F}}(u_{n-1}, u_n) d_{\mathcal{F}}(u_n, u_{n+1})}{1 + d_{\mathcal{F}}(u_{n-1}, u_n)}\}.
\end{aligned}
$$

If $\max\{d_{\mathcal{F}}(u_{n-1}, u_n), \frac{d_{\mathcal{F}}(u_{n-1}, u_n) d_{\mathcal{F}}(u_n, u_{n+1})}{1 + d_{\mathcal{F}}(u_{n-1}, u_n)}\} = \frac{d_{\mathcal{F}}(u_{n-1}, u_n) d_{\mathcal{F}}(u_n, u_{n+1})}{1 + d_{\mathcal{F}}(u_{n-1}, u_n)}$, then, from (5), we obtain

$$d_{\mathcal{F}}(u_n, u_{n+1}) \leq \varphi(\frac{d_{\mathcal{F}}(u_{n-1}, u_n) d_{\mathcal{F}}(u_n, u_{n+1})}{1 + d_{\mathcal{F}}(u_{n-1}, u_n)}) < \frac{d_{\mathcal{F}}(u_{n-1}, u_n) d_{\mathcal{F}}(u_n, u_{n+1})}{1 + d_{\mathcal{F}}(u_{n-1}, u_n)} \leq d_{\mathcal{F}}(u_n, u_{n+1}),$$

which is a contradiction. Hence, $\max\{d_{\mathcal{F}}(u_{n-1}, u_n), \frac{d_{\mathcal{F}}(u_{n-1}, u_n) d_{\mathcal{F}}(u_n, u_{n+1})}{1 + d_{\mathcal{F}}(u_{n-1}, u_n)}\} = d_{\mathcal{F}}(u_{n-1}, u_n)$. Therefore, (5) becomes

$$d_{\mathcal{F}}(u_n, u_{n+1}) \leq \varphi(d_{\mathcal{F}}(u_{n-1}, u_n)). \quad (6)$$

Inductively, we get

$$d_{\mathcal{F}}(u_n, u_{n+1}) \leq \varphi^n(d_{\mathcal{F}}(u_0, u_1)), \quad (7)$$

for all $n \in \mathbb{N}$. Suppose we have $(f, h) \in \mathcal{F} \times [0, +\infty)$ such that $(D_3)$ is assured, and fix $\epsilon > 0$. From $(\mathcal{F}_2)$, $\exists \delta > 0$ such that

$$0 < t < \delta \text{ implies } f(t) < f(\delta) - h. \quad (8)$$

Suppose $n(\epsilon) \in \mathbb{N}$, such that $0 < \sum_{n \geq n(\epsilon)} \varphi^n(d_{\mathcal{F}}(u_0, u_1)) < \delta$. Hence, by (7), $(\mathcal{F}_1)$, and $(\mathcal{F}_2)$, we have

$$f(\sum_{i=n}^{m-1} d_{\mathcal{F}}(u_i, u_{i+1})) \leq f(\sum_{i=n}^{m-1} \varphi^i(d_{\mathcal{F}}(u_0, u_1))) \leq f(\sum_{n \geq n(\epsilon)} \varphi^n(d_{\mathcal{F}}(u_0, u_1))) < f(\epsilon) - h, \quad (9)$$

for $m > n \geq n(\epsilon)$. By $(D_3)$ and (9), we get $d_{\mathcal{F}}(u_n, u_m) > 0$, $m > n \geq n(\epsilon)$ and

$$f(d_{\mathcal{F}}(u_n, u_m)) \leq f(\sum_{i=n}^{m-1} d_{\mathcal{F}}(u_i, u_{i+1})) + h < f(\epsilon),$$

which implies, by $(\mathcal{F}_1)$, that $d_{\mathcal{F}}(\mathfrak{u}_n, \mathfrak{u}_m) < \epsilon$, $m > n \geq n(\epsilon)$, which shows that $\{\mathfrak{u}_n\}$ is $\mathcal{F}$-Cauchy. As $(\mathcal{S}, d_{\mathcal{F}})$ is $\mathcal{F}$-complete, $\exists\, \mathfrak{u}^* \in \mathcal{S}$ such that $\mathfrak{u}_n \to \mathfrak{u}^*$ as $n \to \infty$; that is,

$$\lim_{n \to \infty} d_{\mathcal{F}}(\mathfrak{u}_n, \mathfrak{u}^*) = 0. \tag{10}$$

Suppose that $d_{\mathcal{F}}(\mathcal{J}(\mathfrak{u}^*), \mathfrak{u}^*) > 0$. By $(\mathcal{F}_1)$ and $(D_3)$, we have

$$
\begin{aligned}
f(d_{\mathcal{F}}(\mathcal{J}(\mathfrak{u}^*), \mathfrak{u}^*)) &\leq f(d_{\mathcal{F}}(\mathcal{J}(\mathfrak{u}^*), \mathcal{J}(\mathfrak{u}_n)) + d_{\mathcal{F}}(\mathcal{J}(\mathfrak{u}_n), \mathfrak{u}^*)) + h \\
&\leq f(\alpha(\mathfrak{u}^*, \mathfrak{u}_n) d_{\mathcal{F}}(\mathcal{J}(\mathfrak{u}^*), \mathcal{J}(\mathfrak{u}_n)) + d_{\mathcal{F}}(\mathcal{J}(\mathfrak{u}_n), \mathfrak{u}^*)) + h.
\end{aligned}
$$

By (3), we have

$$
\begin{aligned}
f(d_{\mathcal{F}}(\mathcal{J}(\mathfrak{u}^*), \mathfrak{u}^*)) &\leq f(\alpha(\mathfrak{u}^*, \mathfrak{u}_n) d_{\mathcal{F}}(\mathcal{J}(\mathfrak{u}^*), \mathcal{J}(\mathfrak{u}_n)) + d_{\mathcal{F}}(\mathcal{J}(\mathfrak{u}_n), \mathfrak{u}^*)) + h \\
&\leq f(\alpha(\mathfrak{u}^*, \mathfrak{u}_n) d_{\mathcal{F}}(\mathcal{J}(\mathfrak{u}^*), \mathcal{J}(\mathfrak{u}_n)) + d_{\mathcal{F}}(\mathcal{J}(\mathfrak{u}_n), \mathfrak{u}^*)) + h \\
&\leq f\left(\varphi\left(\max\{d_{\mathcal{F}}(\mathfrak{u}^*, \mathfrak{u}_n), \frac{d_{\mathcal{F}}(\mathfrak{u}^*, \mathcal{J}(\mathfrak{u}^*)) d_{\mathcal{F}}(\mathfrak{u}_n, \mathcal{J}(\mathfrak{u}_n))}{1 + d_{\mathcal{F}}(\mathfrak{u}^*, \mathfrak{u}_n)}\}\right) + d_{\mathcal{F}}(\mathfrak{u}_{n+1}, \mathfrak{u}*)\right) + h, \\
&< f\left(\max\{d_{\mathcal{F}}(\mathfrak{u}^*, \mathfrak{u}_n), \frac{d_{\mathcal{F}}(\mathfrak{u}^*, \mathcal{J}(\mathfrak{u}^*)) d_{\mathcal{F}}(\mathfrak{u}_n, \mathfrak{u}_{n+1})}{1 + d_{\mathcal{F}}(\mathfrak{u}^*, \mathfrak{u}_n)}\} + d_{\mathcal{F}}(\mathfrak{u}_{n+1}, \mathfrak{u}*)\right) + h,
\end{aligned}
$$

for $n \in \mathbb{N}$. If $\max\{d_{\mathcal{F}}(\mathfrak{u}^*, \mathfrak{u}_n), \frac{d_{\mathcal{F}}(\mathfrak{u}^*, \mathcal{J}(\mathfrak{u}^*)) d_{\mathcal{F}}(\mathfrak{u}_n, \mathfrak{u}_{n+1})}{1 + d_{\mathcal{F}}(\mathfrak{u}^*, \mathfrak{u}_n)}\} = d_{\mathcal{F}}(\mathfrak{u}^*, \mathfrak{u}_n)$, then

$$f(d_{\mathcal{F}}(\mathcal{J}(\mathfrak{u}^*), \mathfrak{u}^*)) \leq f(d_{\mathcal{F}}(\mathfrak{u}^*, \mathfrak{u}_n) + d_{\mathcal{F}}(\mathfrak{u}_{n+1}, \mathfrak{u}*)) + h.$$

Taking the limit as $n \to \infty$, and using $(\mathcal{F}_2)$ and (10), we have

$$\lim_{n \to \infty} f(d_{\mathcal{F}}(\mathcal{J}(\mathfrak{u}^*), \mathfrak{u}^*)) \leq \lim_{n \to \infty} f(d_{\mathcal{F}}(\mathfrak{u}^*, \mathfrak{u}_n) + d_{\mathcal{F}}(\mathfrak{u}_{n+1}, \mathfrak{u}*)) + h = -\infty,$$

which implies that $d_{\mathcal{F}}(\mathcal{J}(\mathfrak{u}^*), \mathfrak{u}^*) = 0$, which is a contradiction.

If $\max\{d_{\mathcal{F}}(\mathfrak{u}^*, \mathfrak{u}_n), \frac{d_{\mathcal{F}}(\mathfrak{u}^*, \mathcal{J}(\mathfrak{u}^*)) d_{\mathcal{F}}(\mathfrak{u}_n, \mathfrak{u}_{n+1})}{1 + d_{\mathcal{F}}(\mathfrak{u}^*, \mathfrak{u}_n)}\} = \frac{d_{\mathcal{F}}(\mathfrak{u}^*, \mathcal{J}(\mathfrak{u}^*)) d_{\mathcal{F}}(\mathfrak{u}_n, \mathfrak{u}_{n+1})}{1 + d_{\mathcal{F}}(\mathfrak{u}^*, \mathfrak{u}_n)}$, then

$$f(d_{\mathcal{F}}(\mathcal{J}(\mathfrak{u}^*), \mathfrak{u}^*)) \leq f\left(\frac{d_{\mathcal{F}}(\mathfrak{u}^*, \mathcal{J}(\mathfrak{u}^*)) d_{\mathcal{F}}(\mathfrak{u}_n, \mathfrak{u}_{n+1})}{1 + d_{\mathcal{F}}(\mathfrak{u}^*, \mathfrak{u}_n)} + d_{\mathcal{F}}(\mathfrak{u}_{n+1}, \mathfrak{u}*)\right) + h.$$

Taking the limit as $n \to \infty$, and using $(\mathcal{F}_2)$ and (10), we have

$$\lim_{n \to \infty} f(d_{\mathcal{F}}(\mathcal{J}(\mathfrak{u}^*), \mathfrak{u}^*)) \leq \lim_{n \to \infty} f\left(\frac{d_{\mathcal{F}}(\mathfrak{u}^*, \mathcal{J}(\mathfrak{u}^*)) d_{\mathcal{F}}(\mathfrak{u}_n, \mathfrak{u}_{n+1})}{1 + d_{\mathcal{F}}(\mathfrak{u}^*, \mathfrak{u}_n)} + d_{\mathcal{F}}(\mathfrak{u}_{n+1}, \mathfrak{u}*)\right) + h = -\infty,$$

which implies that $d_{\mathcal{F}}(\mathcal{J}(\mathfrak{u}^*), \mathfrak{u}^*) = 0$, a contradiction. Therefore, we have $d_{\mathcal{F}}(\mathcal{J}(\mathfrak{u}^*), \mathfrak{u}^*) = 0$, i.e. $\mathcal{J}(\mathfrak{u}^*) = \mathfrak{u}^*$. $\quad\square$

Now, we prove that $\mathfrak{u}^*$ is unique. So, we take the following property:
(P) $\alpha(\mathfrak{u}, \mathfrak{v}) \geq 1$ for $\mathcal{J}(\mathfrak{u}) = \mathfrak{u}$ and $\mathcal{J}(\mathfrak{v}) = \mathfrak{v}$ and $\mathfrak{u}, \mathfrak{v} \in \mathcal{S}$.

**Theorem 4.** *Assume the hypotheses of Theorem 3. If we add the property (P), then we get the uniqueness of the fixed point.*

**Proof.** Let $\mathfrak{u}^*, \widehat{\mathfrak{u}} \in \mathcal{S}$ be two fixed points of $\mathcal{J}$ such that $\mathfrak{u}^* \neq \widehat{\mathfrak{u}}$. Then, by hypothesis (P), $\alpha(\mathfrak{u}^*, \widehat{\mathfrak{u}}) \geq 1$. Then,

$$
\begin{aligned}
d_{\mathcal{F}}(\mathfrak{u}^*, \widehat{\mathfrak{u}}) &= d_{\mathcal{F}}(\mathcal{J}(\mathfrak{u}^*), \mathcal{J}(\widehat{\mathfrak{u}})) \leq \alpha(\mathfrak{u}^*, \widehat{\mathfrak{u}}) d_{\mathcal{F}}(\mathcal{J}(\mathfrak{u}^*), \mathcal{J}(\widehat{\mathfrak{u}})) \\
&\leq \varphi(\max\{d_{\mathcal{F}}(\mathfrak{u}^*, \widehat{\mathfrak{u}}), \frac{d_{\mathcal{F}}(\mathfrak{u}^*, \mathcal{J}(\mathfrak{u}^*)) d_{\mathcal{F}}(\widehat{\mathfrak{u}}, \mathcal{J}(\widehat{\mathfrak{u}}))}{1 + d_{\mathcal{F}}(\mathfrak{u}^*, \widehat{\mathfrak{u}})}\}) \\
&= \varphi(d_{\mathcal{F}}(\mathfrak{u}^*, \widehat{\mathfrak{u}})) \\
&< d_{\mathcal{F}}(\mathfrak{u}^*, \widehat{\mathfrak{u}}),
\end{aligned}
$$

which is a contradiction. Hence, $\mathcal{J}$ has a unique fixed point in $\mathcal{S}$. $\square$

**Example 3.** *Let $\mathcal{S} = \mathbb{R}$ and $d_{\mathcal{F}}$ be an $\mathcal{F}$-metric given by*

$$
d_{\mathcal{F}}(\mathfrak{u}, \mathfrak{v}) = \begin{cases} e^{|\mathfrak{u}-\mathfrak{v}|}, & if\ \mathfrak{u} \neq \mathfrak{v} \\ 0, & if\ \mathfrak{u} = \mathfrak{v}. \end{cases}
$$

*Take $f(t) = \frac{-1}{t}$ and $h = 1$. Define $\mathcal{J} : \mathcal{S} \rightarrow \mathcal{S}$ by*

$$
\mathcal{J}(\mathfrak{u}) = \begin{cases} 3\mathfrak{u}, & if\ \mathfrak{u} > 1 \\ \frac{\mathfrak{u}}{3}, & if\ 0 \leq \mathfrak{u} \leq 1 \\ 0, & otherwise. \end{cases}
$$

*Now we define $\alpha : \mathcal{S} \times \mathcal{S} \rightarrow [0, +\infty)$ by*

$$
\alpha(\mathfrak{u}, \mathfrak{v}) = \begin{cases} 1 & if\ \mathfrak{u}, \mathfrak{v} \in [0, 1] \\ 0, & otherwise. \end{cases}
$$

*Clearly, $\mathcal{J}$ is an $(\alpha, \varphi)$ rational contraction with $\varphi(t) = kt$ for all $t \geq 0$ and $k \in (0, 1)$. In fact, for all $\mathfrak{u}, \mathfrak{v} \in \mathcal{S}$, we have*

$$
d_{\mathcal{F}}(\mathcal{J}(\mathfrak{u}), \mathcal{J}(\mathfrak{v})) \leq k\left(\max\{d_{\mathcal{F}}(\mathfrak{u}, \mathfrak{v}), \frac{d_{\mathcal{F}}(\mathfrak{u}, \mathcal{J}(\mathfrak{u})) d_{\mathcal{F}}(\mathfrak{v}, \mathcal{J}(\mathfrak{v}))}{1 + d_{\mathcal{F}}(\mathfrak{u}, \mathfrak{v})}\}\right).
$$

*All the conditions of Theorem 3 are satisfied and, hence, there exists a unique $0 \in \mathcal{S}$, such that $\mathcal{J}(0) = 0$.*

### 3.2. Fixed Point Result for Cyclic Contractions

Another attractive topic in fixed point theory is the concept of cyclic mappings, introduced by Kirk et al. [9] in 2003. Later on, Shahzad et al. [10,11] utilized this notion and obtained some fixed and proximity point results in complete metric spaces. In this section, we define a cyclic contraction in the context of an $\mathcal{F}$-metric space, as follows:

**Definition 5.** *Let $\mathcal{S}$ be a non-empty set, $m$ be a positive integer, and $\mathcal{J} : \mathcal{S} \rightarrow \mathcal{S}$ be an operator. By definition, $\mathcal{S} = \cup_{i=1}^{m} \mathcal{S}_i$ is a cyclic representation of $\mathcal{S}$ with respect to $\mathcal{J}$, if*

(1) $\mathcal{S}_i, i = 1, 2, ..., m$ *are non-empty sets, and*
(2) $\mathcal{J}(\mathcal{S}_1) \subseteq \mathcal{S}_2, \mathcal{J}(\mathcal{S}_2) \subseteq \mathcal{S}_3, ..., \mathcal{J}(\mathcal{S}_{m-1}) \subseteq \mathcal{S}_m, \mathcal{J}(\mathcal{S}_m) \subseteq \mathcal{S}_1$.

**Definition 6.** *Let $(\mathcal{S}, d_{\mathcal{F}})$ be an $\mathcal{F}$-metric space and $\{A_j\}_{j=1}^{m}$ be a family of non-empty closed subsets of $\mathcal{S}$ and $Y = \cup_{j=1}^{m} A_j$. A self-mapping $\mathcal{J} : Y \rightarrow Y$ is said to be a cyclic contraction if*

$$
\mathcal{J}(A_j) \subseteq A_{j+1}, j = 1, 2, ..., m, where\ A_{m+1} = A_1,
$$

*and*

$$d_{\mathcal{F}}(\mathcal{J}(\mathfrak{u}), \mathcal{J}(\mathfrak{v})) \leq \lambda d_{\mathcal{F}}(\mathfrak{u}, \mathfrak{v}), \tag{11}$$

*for all $\mathfrak{u} \in A_j$ and $\mathfrak{v} \in A_{j+1}$, $j = 1, 2, ..., m$, where $\lambda \in (0, 1)$.*

**Theorem 5.** *Let $(\mathcal{S}, d_{\mathcal{F}})$ be a complete $\mathcal{F}$-metric space and $\mathcal{J} : Y \to Y$ be a cyclic contraction. Then, $\mathcal{J}$ has a unique fixed point in $\cap_{j=1}^{m} A_j$.*

**Proof.** Let $\mathfrak{u}_0 \in Y$ be an arbitrary element. Without loss of generality, we assume that $\mathfrak{u}_0 \in A_1$. Define the sequence $\mathfrak{u}_{n+1} = \mathcal{J}\mathfrak{u}_n$ for all $n \in \mathbb{N}$. As $\mathcal{J}$ is cyclic, $\mathfrak{u}_0 \in A_1$, $\mathfrak{u}_1 = \mathcal{J}\mathfrak{u}_0 \in A_2$, $\mathfrak{u}_2 = \mathcal{J}\mathfrak{u}_1 \in A_3, ...$, and so on. If $\mathfrak{u}_{n_0+1} = \mathfrak{u}_{n_0}$ for some $n_0 \in \mathbb{N}$, then, obviously, the fixed point of $\mathcal{J}$ is $\mathfrak{u}_{n_0}$. So, we assume that $\mathfrak{u}_{n+1} \neq \mathfrak{u}_n$ for all $n \in \mathbb{N}$. Then, by (11), we have

$$d_{\mathcal{F}}(\mathfrak{u}_n, \mathfrak{u}_{n+1}) = d_{\mathcal{F}}(\mathcal{J}\mathfrak{u}_{n-1}, \mathcal{J}\mathfrak{u}_n) \leq \lambda d_{\mathcal{F}}(\mathfrak{u}_{n-1}, \mathfrak{u}_n) \leq \lambda^2 d_{\mathcal{F}}(\mathfrak{u}_{n-2}, \mathfrak{u}_{n-1}) \leq ... \leq \lambda^n d_{\mathcal{F}}(\mathfrak{u}_0, \mathfrak{u}_1), \tag{12}$$

for $n \in \mathbb{N}$, which implies that

$$\sum_{i=n}^{m-1} d_{\mathcal{F}}(\mathfrak{u}_i, \mathfrak{u}_{i+1}) \leq \frac{\lambda^n}{1-\lambda} d_{\mathcal{F}}(\mathfrak{u}_0, \mathfrak{u}_1), \quad m > n. \tag{13}$$

As

$$\lim_{n \to \infty} \frac{\lambda^n}{1-\lambda} d_{\mathcal{F}}(\mathfrak{u}_0, \mathfrak{u}_1) = 0,$$

there $\exists N \in \mathbb{N}$ such that

$$0 < \frac{\lambda^n}{1-\lambda} d_{\mathcal{F}}(\mathfrak{u}_0, \mathfrak{u}_1) < \delta, \ n \geq N. \tag{14}$$

Hence, by (14) and $(\mathcal{F}_2)$, we get

$$f\left(\sum_{i=n}^{m-1} d_{\mathcal{F}}(\mathfrak{u}_i, \mathfrak{u}_{i+1})\right) \leq f\left(\frac{\lambda^n}{1-\lambda} d_{\mathcal{F}}(\mathfrak{u}_0, \mathfrak{u}_1)\right) < f(\epsilon) - h, \tag{15}$$

for $m > n \geq N$. Applying $(D_3)$ and (15), we get $d_{\mathcal{F}}(\mathfrak{u}_n, \mathfrak{u}_m) > 0$, $m > n \geq N$, such that

$$f(d_{\mathcal{F}}(\mathfrak{u}_n, \mathfrak{u}_m)) \leq f\left(\sum_{i=n}^{m-1} d_{\mathcal{F}}(\mathfrak{u}_i, \mathfrak{u}_{i+1})\right) + h < f(\epsilon),$$

which implies, by $(\mathcal{F}_1)$, that $d_{\mathcal{F}}(\mathfrak{u}_n, \mathfrak{u}_m) < \epsilon$, $m > n \geq N$, which demonstrates that $\{\mathfrak{u}_n\}$ is $\mathcal{F}$-Cauchy. Now, the completeness of $\mathcal{S}$ implies that there exists $\mathfrak{u}^* \in \mathcal{S}$, such that

$$\lim_{n \to \infty} d_{\mathcal{F}}(\mathfrak{u}_n, \mathfrak{u}^*) = 0. \tag{16}$$

It is easy to see that $\mathfrak{u}^* \in \cap_{j=1}^{m} A_j$. Indeed, if $\mathfrak{u}_0 \in A_1$, then $\{\mathfrak{u}_{m(n-1)}\}_{n=1}^{\infty} \in A_1$ and $\{\mathfrak{u}_{m(n-1)+1}\}_{n=1}^{\infty} \in A_2$. Pursuing in this way, we have $\{\mathfrak{u}_{mn-1}\}_{n=1}^{\infty} \in A_m$. All of these subsequences are convergent. They all converge to the one point $\mathfrak{u}^*$. Furthermore, the sets $A_j$ are closed. Hence,

$$\mathfrak{u}^* \in \cap_{j=1}^{m} A_j. \tag{17}$$

Now, we prove that $\mathfrak{u}^*$ is a fixed point of $\mathcal{J}$. Assume, on the contrary, that $\mathcal{J}\mathfrak{u}^* \neq \mathfrak{u}^*$. Then, $d_{\mathcal{F}}(\mathcal{J}\mathfrak{u}^*, \mathfrak{u}^*) > 0$. By $(D_3)$, we have

$$
\begin{aligned}
f(d_{\mathcal{F}}(\mathcal{J}(\mathfrak{u}^*), \mathfrak{u}^*)) &\leq f(d_{\mathcal{F}}(\mathcal{J}(\mathfrak{u}^*), \mathfrak{u}_{n+1}) + d_{\mathcal{F}}(\mathfrak{u}_{n+1}, \mathfrak{u}^*)) \\
&\leq f(d_{\mathcal{F}}(\mathcal{J}(\mathfrak{u}^*), \mathcal{J}(\mathfrak{u}_n)) + d_{\mathcal{F}}(\mathfrak{u}_{n+1}, \mathfrak{u}^*)) \\
&\leq f(\lambda d_{\mathcal{F}}(\mathfrak{u}^*, \mathfrak{u}_n) + d_{\mathcal{F}}(\mathfrak{u}_{n+1}, \mathfrak{u}^*)) + h.
\end{aligned}
$$

Letting $n \to \infty$, we get

$$f(d_{\mathcal{F}}(\mathcal{J}(\mathfrak{u}^*), \mathfrak{u}^*)) \leq \lim_{n \to \infty} f(\lambda d_{\mathcal{F}}(\mathfrak{u}^*, \mathfrak{u}_n) + d_{\mathcal{F}}(\mathfrak{u}_{n+1}, \mathfrak{u}^*)) + h.$$

By $(\mathcal{F}_2)$ and (17), we have

$$\lim_{n \to \infty} f(\lambda d_{\mathcal{F}}(\mathfrak{u}^*, \mathfrak{u}_n) + d_{\mathcal{F}}(\mathfrak{u}_{n+1}, \mathfrak{u}^*)) + h = -\infty.$$

This implies that $d_{\mathcal{F}}(\mathcal{J}(\mathfrak{u}^*), \mathfrak{u}^*) = 0$, which is a contradiction. Thus, $\mathcal{J}(\mathfrak{u}^*) = \mathfrak{u}^*$. Now, we show that $\mathfrak{u}^*$ is unique. Assume, on the contrary, that there exist two distinct fixed points $\mathfrak{u}^*$ and $\widehat{\mathfrak{u}}$ of $\mathcal{J}$; that is, $\mathcal{J}(\mathfrak{u}^*) = \mathfrak{u}^*$, $\mathcal{J}(\widehat{\mathfrak{u}}) = \widehat{\mathfrak{u}}$, and $\mathfrak{u}^* \neq \widehat{\mathfrak{u}}$. Then, $d_{\mathcal{F}}(\mathfrak{u}^*, \widehat{\mathfrak{u}}) > 0$. Now, by definition, we have

$$d_{\mathcal{F}}(\mathfrak{u}^*, \widehat{\mathfrak{u}}) = d_{\mathcal{F}}(\mathcal{J}(\mathfrak{u}^*), \mathcal{J}(\widehat{\mathfrak{u}})) \leq \lambda d_{\mathcal{F}}(\mathfrak{u}^*, \widehat{\mathfrak{u}}) < d_{\mathcal{F}}(\mathfrak{u}^*, \widehat{\mathfrak{u}}),$$

which is a contradiction. Thus, $\mathfrak{u}^* = \widehat{\mathfrak{u}}$. □

*3.3. Applications*

In this section, we will discuss the solution of the following differential equation

$$u'(t) = -e_1(t)u(t) + e_2(t)g(u(t - s(t))) + e_3(t)u'(t - s(t)). \tag{18}$$

The following lemma, of Djoudi et al. [12], will prove to be very useful.

**Lemma 2.** *[12] Assume that $s'(t) \neq 1 \ \forall t \in \mathbb{R}$. Then, $u(t)$ is a solution of (18) if*

$$u(t) = \left( u(0) - \frac{e_3(0)}{1 - s'(0)} u(-s(0)) \right) e^{-\int_0^t \alpha(s)ds} + \frac{e_3(t)}{1 - s'(t)} u(t - s(t))$$

$$- \int_0^t (h(w))u(w - s(w))) - e_2(w)g(u(w - s(w)))) e^{-\int_w^t \alpha(s)ds} dw, \tag{19}$$

*where*

$$h(w) = \frac{s''(w)e_3(w) + \left( e_3'(w) + e_3(w)e_1(w) \right)(1 - s'(w))}{(1 - s'(w))^2}. \tag{20}$$

*Now, suppose that $\varphi : (-\infty, 0] \to \mathbb{R}$ is a continuous bounded initial function. Then, $u(t) = u(t, 0, \varphi)$ is a solution of (18) if $u(t) = \varphi(t)$ for $t \leq 0$ and assures (18) for $t \geq 0$. Let $\mathfrak{C}$ be the space of all continuous functions from $\mathbb{R}$ to $\mathbb{R}$. Define the set $\mathcal{B}_\varphi$ by*

$$\mathcal{B}_\varphi = \{ \tau : \mathbb{R} \to \mathbb{R} \text{ such that } \varphi(t) = \tau(t) \text{ if } t \leq 0, \ \tau(t) \to 0 \text{ as } t \to \infty, \ \tau \in \mathfrak{C} \}.$$

*Then, $\mathcal{B}_\varphi$ is a Banach space equipped with the supremum norm $\|\cdot\|$.*

**Lemma 3.** *[13] The space $(\mathcal{B}_\varphi, \|\cdot\|)$ provided with $d$ given by*

$$d(\mathfrak{t}, \mathfrak{t}^*) = \|\mathfrak{t} - \mathfrak{t}^*\| = \sup_{u \in I} |\mathfrak{t}(u) - \mathfrak{t}^*(u)|,$$

*for $\mathfrak{t}, \mathfrak{t}^* \in \mathcal{B}_\varphi$, is an $\mathcal{F}$-metric space.*

We state and prove the followin theorem as an application of our main result.

**Theorem 6.** *Let $Q : \mathcal{B}_\varphi \to \mathcal{B}_\varphi$ be the mapping defined by*

$$(Q\tau)(t) = \left( \tau(0) - \frac{e_3(0)}{1 - s'(0)} \tau(-s(0)) \right) e^{-\int_0^t \alpha(s)ds} + \frac{e_3(t)}{1 - s'(t)} \tau(t - s(t))$$

$$- \int_0^t (h(w)\tau(w - s(w)) - e_2(w)g\left(\tau(w - s(w))\right))e^{-\int_w^t \alpha(s)ds}dw, \; t \geq 0 \qquad (21)$$

*for all $\tau \in \mathcal{B}_\varphi$. Assume that these assertions are satisfied:*

(i) *There exist $\mu \geq 0$ and $\varphi \in \Psi$ such that*

$$\int_0^t |h(w)(\tau(w - s(w))) - \sigma(w - s(w))|e^{-\int_w^t \alpha(s)ds}$$

$$\leq \frac{\mu}{2}\varphi\left( \max\left\{ ||\tau - \sigma||, \frac{||\tau - Q(\tau)||\,||\sigma - Q(\sigma)||}{1 + ||\tau - \sigma||} \right\} \right) \qquad (22)$$

*and*

$$\int_0^t |(e_2(w))g(\tau(w - s(w))) - g(\sigma(w - s(w)))|e^{-\int_w^t \alpha(s)ds}$$

$$\leq \frac{\mu}{2}\varphi\left( \max\left\{ ||\tau - \sigma||, \frac{||\tau - Q(\tau)||\,||\sigma - Q(\sigma)||}{1 + ||\tau - \sigma||} \right\} \right) \qquad (23)$$

*for all $\tau, \sigma \in \mathcal{B}_\varphi$; and*

(ii)

$$\left| \frac{e_3(t)}{1 - s'(t)} \right| + \mu \leq 1, \quad t \geq 0. \qquad (24)$$

*Then, $Q$ has a fixed point.*

**Proof.** Define $\alpha : \mathfrak{C} \times \mathfrak{C} \to \mathbb{R}$ by

$$\alpha(\tau, \sigma) = \begin{cases} 1, & \text{if } \tau, \sigma \in \mathcal{B}_\varphi, \\ 0, & \text{otherwise.} \end{cases}$$

Now, let $\tau, \sigma \in \mathcal{B}_\varphi$ such that $\alpha(\tau, \sigma) \geq 1$. It follows, from (21), that $Q(\tau), Q(\sigma) \in \mathcal{B}_\varphi$. Therefore, $\alpha(Q(\tau), Q(\sigma)) \geq 1$. As (22)–(24) hold, then, for $\tau, \sigma \in \mathcal{B}_\varphi$, we have

$$
\begin{aligned}
|(Q\tau)(t) - (Q\sigma)(t)| \; &\leq \; \left| \frac{e_3(t)}{1 - s'(t)} \right| ||\tau - \sigma|| \\
&\quad + \int_0^t |h(w)(\tau(w - s(w))) - \sigma(w - s(w))|e^{-\int_w^t \alpha(s)ds} \\
&\quad \int_0^t |(e_2(w))g(\tau(w - s(w))) - g(\sigma(w - s(w)))|e^{-\int_w^t \alpha(s)ds} \\
&\leq \; \left| \frac{e_3(t)}{1 - s'(t)} \right| ||\tau - \sigma|| + \mu\varphi\left( \max\left\{ ||\tau - \sigma||, \frac{||\tau - Q\tau||\,||\sigma - Q\sigma||}{1 + ||\tau - \sigma||} \right\} \right) \\
&\leq \; \left\{ \left| \frac{e_3(t)}{1 - s'(t)} \right| + \mu \right\} \varphi\left( \max\left\{ ||\tau - \sigma||, \frac{||\tau - Q\tau||\,||\sigma - Q\sigma||}{1 + ||\tau - \sigma||} \right\} \right) \\
&\leq \; \varphi\left( \max\left\{ ||\tau - \sigma||, \frac{||\tau - Q\tau||\,||\sigma - Q\sigma||}{1 + ||\tau - \sigma||} \right\} \right).
\end{aligned}
$$

As $R(\tau, \sigma) = \max\left\{ ||\tau - \sigma||, \frac{||\tau - Q\tau||\,||\sigma - Q\sigma||}{1 + ||\tau - \sigma||} \right\}$, we have

$$|(Q\tau)(t) - (Q\sigma)(t)| \leq \varphi(R(\tau, \sigma)).$$

Hence,

$$\alpha(\tau, \sigma)d(Q\tau, Q\sigma) = d(Q\tau, Q\sigma) \leq \varphi(R(\tau, \sigma)),$$

which implies that $Q$ is a rational $(\alpha, \varphi)$-contraction. Thus, by Theorem 3, $Q$ has a unique fixed point in $\mathcal{B}_\varphi$ which solves (18). □

## 4. Discussion

To generalize the notion of a metric space, a new class of metric spaces, called $\mathcal{F}$-metric spaces, was introduced by Jleli and Samet [5]. It was shown that any standard metric $d$ is an $\mathcal{F}$-metric $d_{\mathcal{F}}$; however, the converse is not true. Actually, if $d_{\mathcal{F}}$ is a metric on $\mathcal{S}$, then the conditions $(D_1)$ and $(D_2)$ are satisfied. Otherwise, by the triangle inequality, for every $(\mathfrak{u}, \mathfrak{v}) \in \mathcal{S} \times \mathcal{S}$, for each $N \in \mathbb{N}$ ($N \geq 2$), and for every $(\mathfrak{w}_i)_{i=1}^N \subset \mathcal{S}$ such that $(\mathfrak{w}_1, \mathfrak{w}_N) = (\mathfrak{u}, \mathfrak{v})$, we have

$$d_{\mathcal{F}}(\mathfrak{u}, \mathfrak{v}) \leq \sum_{i=1}^{N-1} d_{\mathcal{F}}(\mathfrak{u}_i, \mathfrak{u}_{i+1}),$$

which implies that

$$d_{\mathcal{F}}(\mathfrak{u}, \mathfrak{v}) > 0 \Rightarrow \ln(d_{\mathcal{F}}(\mathfrak{u}, \mathfrak{v})) \leq \ln\left(\sum_{i=1}^{N-1} d_{\mathcal{F}}(\mathfrak{u}_i, \mathfrak{u}_{i+1})\right).$$

Then, $d_{\mathcal{F}}$ assures $(D_3)$ with $f(\theta) = \ln(\theta)$, $\theta > 0$, and $h = 0$. In this paper, some fixed point theorems for $(\alpha, \varphi)$ rational contractions and cyclic contractions, in the context of $\mathcal{F}$-metric spaces, are established. By the above note, our main Theorems 3 and 5 are real generalizations of the results of [9]. By example 1, several fixed point theorems can be obtained in $\mathcal{F}$-metric spaces.

## 5. Conclusions

In the present paper, we have defined $(\alpha, \varphi)$ rational contractions and cyclic contractions in the setting of $\mathcal{F}$-metric spaces and obatined some generalized fixed point results. The neutral delay differential equations seen in the modelling of networks involving lossless transmission lines and in investigations of vibrating masses attached to an elastic bar, as well as used as the Euler equation in some variational problems, theory of automatic control, and neuromechanical systems in which inertia plays a significant role. As an application of our main results, the existence of solution for a certain differential equation is also investigated. We also have provided an example to support the new theorem. Our results are new and significantly contribute to the existing literature in the fixed point theory.

In this area, our future work will focus on studying the fixed points of multi-valued and fuzzy mappings in $\mathcal{F}$-metric spaces, with fractional differential inclusion problems as applications.

**Author Contributions:** All authors contributed equally and significantly in writing this paper. All authors read and approved the final paper.

**Funding:** This project was funded by Deanship of Scientific Research (DSR), Taibah University, Al Madina Al Munawara, Kingdom of Saudi Arabia, under Grant No. 60348/1439.

**Acknowledgments:** Authors are very much thankful to the referees for their careful reading of the manuscript and suggestions. The comments of the referees were very useful and they helped us to improve the paper significantly. The authors are thankful to DSR for providing research facilities and financial support.

**Conflicts of Interest:** The authors declare that they have no competing interest.

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
