# Peer review of "New Fixed Point Theorems with Applications to Non-Linear Neutral Differential Equations"

_symmetry, doi:10.3390/sym11050602_

Round 1
Reviewer 1 Report
It is a very well-structured article. There are some issues to be delt with:
-The authors state with the manuscript title “New Fixed Point Theorems” and in the abstract “we take advantage of fixed point theorems for self mappings satisfying a new (a, j) rational contraction as well as cyclic contraction in the context of F-metric spaces”. The authors should clarify and discuss the novelty of their study. It is not clear in the manuscript which are the new elements they propose. The word “new” is found only in the title (New Fixed Point Theorems) and in the abstract (new (a, j) rational contraction as well as cyclic contraction).
-The authors refer “Theorem 4. With hypotheses of Theorem 2.1, if we add the property (P)” but I cannot find Theorem 2.1 anywhere in the manuscript.
-There is no conclusion-discussion section.
-The manuscript has many similar parts with the article:
Djoudi, A., & Khemis, R. (2006). Fixed point techniques and stability for neutral nonlinear differential equations with unbounded delays. Georgian Mathematical Journal, 13(1), 25-34. http://www.heldermann-verlag.de/gmj/gmj13/gmj13004.pdf
Section “4. Applications” of the article: Pages 6-9 are almost the same with the above article.
Author Response
Dear Reviewer
Please find the manuscript with point-by point response.
We have improved the introduction by adding some suitable results and beginning note in sections 1, 2 and 3.
We have corrected the sentence
“With hypotheses of Theorem 2.1, if we add the property (P)” and it is given properly.
The given two sections (1 and 2) are actually “Results and Discussion sections”.
We have added the “Conclusions” section which summarizes our investigations and future work in this direction. We also highlighted some applications of the investigated results in real world problems.
Moreover, we have established our results in F-metric space which is introduced first time in 2018.
“Every metric space is F-metric space but the converse is not true.”
Secondly the contractive condition “
α(u,v)d_{F}(J(u),J(v))≤ϕ(R(u,v))
where
R(u,v)=max{d_{F}(u,v),((d_{F}(u,J(u))d_{F}(v,J(v)))/(1+d_{F}(u,v)))}
“given in our paper is more general than the contractive condition in
Djoudi, A., & Khemis, R. [Georgian Mathematical Journal, 13(1)(2006), 25-34]
so our results in the context of F-metric space are new\generalized.
We overall improved the quality of the paper.
We all are very much thankful to you for your careful reading of the manuscript and suggestions.
With Best Wishes
Dr. Jamshaid Ahmad
Reviewer 2 Report
I have reviewed the manuscript "New Fixed Point Theorems with Applications to Nonlinear Neutral Differential Equations", Manuscript ID: symmetry-451449 that has been submitted for publication in the MDPI Symmetry Journal and I have identified a series of aspects that in my opinion must be addressed in order to bring a benefit to the manuscript. In this paper, the authors study the existence of a solution for a nonlinear neutral differential equation with an unbounded delay, using fixed point theorems. First of all, I consider that the authors should improve their presentation, highlighting the originality and novelty of their approach in rapport with previous studies. Secondly, I consider that the article will benefit if the authors take into account the following remarks and address within the manuscript the signaled issues:
1) The line numbers are missing in the whole pdf manuscript submitted by the authors, so I had to refer to excerpts of text and page numbers within the comments.
2) The sections of the manuscript. The manuscript under review will benefit if it is restructured in accordance with the Symmetry MDPI Journal's Template that provides a more logical structure that is much more appropriate for a research article. The restructuring of the manuscript will also help the authors to better express the novelty of their work and the contribution that they have made to the current state of knowledge. Consequently, the manuscript under review should be restructured as follows: Abstract, Keywords, 1. Introduction, 2. Materials and Methods, 3. Results, 4. Discussion, 5. Conclusions (not mandatory), 6. Patents (not mandatory), Supplementary Materials (not mandatory), Author Contributions, Funding, Acknowledgments, Conflicts of Interest, Appendices and References. Moreover, the authors must take into account the recommendations from the MDPI Symmetry Journal website regarding the format of the papers, by using the Microsoft Word template or LaTeX template to prepare their manuscript. In the actual form of the paper, most of these recommendations have not been taken into account and thus the reading of the manuscript is affected.
3) The "Introduction" section. In the "Introduction" section, the authors have presented the literature review of the cited papers in the following manner: "Afterward, Alnaser et al. [7] obtained the relation theoretic contraction results owing the concept of F-metric space. For more details, we refer the reader to [1-34]." I consider that the literature review should be improved by performing a careful analysis of the cited works. The authors must highlight exactly, for each of the involved referenced papers the main contribution that the authors of the referenced papers have brought to the current state of knowledge, the method used by the authors of the referenced papers, a brief presentation of the main obtained results and some limitations of the referenced article. This is the only way to contextualize the current state of the art in which the authors of the manuscript position their paper, identify and address aspects that have not been tackled/solved yet by the existing studies.
4) The "Introduction" section. In the "Introduction" section, the authors must state more clearly a gap in the current state of knowledge that needs to be filled, a gap that is being addressed by their manuscript. At the end of the Introduction, the authors should present the structure of their paper, under the form: "The rest of the paper is structured as follows: Section 2 contains…".
5) The "Introduction and preliminaries" section. On Page 1, at the condition F2, the authors refer first to a sequence tn and afterwards to a sequence αn that has not been previously defined. For consistency reasons, please verify this condition.
6) The "Introduction and preliminaries" section. On page 2, in example 1, "…with an F-metric dF define by…". Please verify this sentence, in the current form its meaning is not clear. Probably the authors intended to write "…with an F-metric dF defined by…".
7) The "Materials and Methods" section. In the actual form of the manuscript, the "Materials and Methods" section is missing. It will benefit the paper if the authors restructure their paper and devise a proper "Materials and Methods" section, as requested by the Symmetry MDPI Journal's Template. I consider that the authors must pay more attention to the appropriate citation of the methods and results that have been retrieved from the scientific literature. When the authors present the information in the "Materials and Methods" section, they must assume clearly their original contribution by specifying this fact and by highlighting the fact that starting from a certain point there are depicted the original and novel aspects of their research. When presenting the devised method/approach in the "Materials and Methods" section, it will be extremely helpful to design and insert a diagram depicting the main sequence of steps that one has to process in order to reproduce the results of the conducted study. This diagram should be analyzed in detail within the manuscript by specifying all the elements needed for each and every step, in order to reach the final result of the study.
8) The "Discussion" section. In order to validate the usefulness of their research, in the "Discussion" section that is currently missing from the manuscript, the authors should make a detailed comparison between their study from the manuscript and other ones that have been developed and used in the literature for the same or similar purposes. In the "Discussion" section the authors should also highlight current limitations of their study, and briefly mention some precise directions that they intend to follow in their future research work. The paper will benefit if the authors make a step further, beyond their approach and provide an insight at the end of the "Discussion" section regarding what they consider to be, based on the obtained results, the most important steps that all the involved parties should take in order to benefit from the results of the research conducted within the manuscript.
9) The "Conclusions" section. Although the "Conclusions" section is optional, I consider that the results are insufficiently explained and interpreted if the manuscript is concluded with the sentence "Thus by Theorem 3, Q has a unique fixed point in B which solves (4.1).", like the authors did. It will benefit the manuscript if the authors devise a proper "Conclusions" section in which they state the most important outcome of their work. The authors should avoid simply summarizing the aspects that they have already stated in the body of the manuscript. Instead, they should interpret their findings at a higher level of abstraction than in the Discussion section. The authors should highlight whether, or to what extent they have managed to address the target proposed within the Introduction section. The authors should avoid restating everything they did once again, but instead they should emphasize what their findings actually mean to the readers, therefore making the Conclusions section interesting and memorable to them.
10) The References within the paper. The references are not cited in an ascending order (for example, the first reference is [19], being followed by [26], [7], [1-34], [17] and [20]). According to the Symmetry MDPI Journal's Template, the references must be numbered in the order of their appearance in the text (including citations in tables and legends) and listed individually at the end of the manuscript. In the actual form of the paper, the references are ordered in an alphabetical order instead of the recommended one. Please reorder the references in the References section, according to the recommendations. Moreover, the reference [1] does not even exist in the References section, it is empty, while reference [35] from the "References" section has not been cited in the manuscript at all. Please address this issue either by citing this paper in the manuscript if it has been used in documenting the paper, or by deleting it in the opposite case.
Author Response
Dear Reviewer
Symmetry Journal
Title: New Fixed Point Theorems with Applications to Nonlinear Neutral Differential Equations
Please find the manuscript with point-by point response.
We have improved the introduction by adding some suitable results and beginning note in sections 1, 2 and 3.
We have corrected (F2) that is
For every sequence {α_{n}}⊆R⁺, lim_{n→∞}α_{n}=0 if and only if lim_{n→∞}f(α_{n})=-∞.
We have corrected the mistake.
The space (ℝ,d_{F}) is an F-metric with an F-metric d_{F} defined by.
The given two sections are actually “Results and Discussion sections”.
We have added the “Conclusions” section which summarizes our investigations and future work in this direction. We also highlighted some applications of the investigated results in real world problems.
We have improved the “Reference” section. We have deleted the extra references. Now all the references are cited properly.
We overall improved the quality of the paper.
We all are very much thankful to you for your careful reading of the manuscript and suggestions.
With Best Wishes
Dr. Jamshaid Ahmad
Round 2
Reviewer 1 Report
The manuscript has been improved. Although pages 6-9 of the article has many similar parts with the article: Djoudi, A., & Khemis, R. (2006). Fixed point techniques and stability for neutral nonlinear differential equations with unbounded delays. Georgian Mathematical Journal, 13(1), 25-34. http://www.heldermann-verlag.de/gmj/gmj13/gmj13004.pdf
Author Response
Title: Fixed Point Theorems with Applications to Nonlinear Neutral Differential Equations
Authors: Jamshaid Ahmad, Laila A. Alnaser, Durdana Lateef, Hoda A. Fouad
Comments: The manuscript has been improved. Although pages 6-9 of the article has many similar parts with the article: Djoudi, A., & Khemis, R. (2006). Fixed point techniques and stability for neutral nonlinear differential equations with unbounded delays. Georgian Mathematical Journal, 13(1), 25-34. http://www.heldermann-verlag.de/gmj/gmj13/gmj13004.pdf
Reply:
Our result is in F-metric space with generalized rational contraction, while the result of Djoudi & Khemis is in metric space. We want to state the following remark here.
This notion of F-metric is more comprehensive than the notion of standard metric because any metric d is an F-metric d_{F} but the converse is not true. By this remark, our result is more general and broad.
Moreover, we have the following contractive condition
α(u,v)d_{F}(J(u),J(v))≤ϕ(R(u,v))
where
R(u,v)=max{d_{F}(u,v),((d_{F}(u,J(u))d_{F}(v,J(v)))/(1+d_{F}(u,v)))}
which is more general because if f(t)=ln(t), h=0,α(u,v)=1 and
max{d_{F}(u,v),((d_{F}(u,J(u))d_{F}(v,J(v)))/(1+d_{F}(u,v)))}=d_{F}(u,v),
then above mentioned result of Djoudi & Khemis can be derived from our main result.
Moreover, we have impoved our paper again.
We are very grateful to you for your positive and helpful suggestions and we feel that the quality of the manuscript has been significantly improved as a result.
Reviewer 2 Report
I have reviewed the revised version of the manuscript "New Fixed Point Theorems with Applications to Nonlinear Neutral Differential Equations", Manuscript ID: symmetry-451449 and, although the authors have modified the paper, there are still a lot of issues that have not been addressed even if I have signaled them in my previous review report. Consequently, I have made and sent to the authors the following comments:
The authors have not provided a proper coverletter to my review report and a point-by-point response to my previous review report, while the majority of the signaled issues have not been addressed. In addition to these, the changes made within the manuscript have not been highlighted or track-changed.
I consider that the authors should improve their presentation, highlighting the originality and novelty of their approach in rapport with previous studies. The authors did not address this issue in the revised version of the manuscript even if I have signaled it in my previous review report.
I consider that the article will benefit if the authors take into account the following remarks and address within the manuscript the signaled issues:
1) The line numbers are missing in the whole pdf manuscript submitted by the authors, so I had to refer to excerpts of text and page numbers within the comments. The authors did not address this issue in the revised version of the manuscript even if I have signaled it in my previous review report.
2) The sections of the manuscript. The manuscript under review will benefit if it is restructured in accordance with the Symmetry MDPI Journal's Template that provides a more logical structure that is much more appropriate for a research article. The restructuring of the manuscript will also help the authors to better express the novelty of their work and the contribution that they have made to the current state of knowledge. Consequently, the manuscript under review should be restructured as follows: Abstract, Keywords, 1. Introduction, 2. Materials and Methods, 3. Results, 4. Discussion, 5. Conclusions (not mandatory), 6. Patents (not mandatory), Supplementary Materials (not mandatory), Author Contributions, Funding, Acknowledgments, Conflicts of Interest, Appendices and References. Moreover, the authors must take into account the recommendations from the MDPI Symmetry Journal website regarding the format of the papers, by using the Microsoft Word template or LaTeX template to prepare their manuscript. In the actual form of the paper, most of these recommendations have not been taken into account and thus the reading of the manuscript is affected. The authors did not address this issue in the revised version of the manuscript even if I have signaled it in my previous review report.
3) The "Introduction" section. I consider that the literature review should be improved by performing a careful analysis of the cited works. The authors must highlight exactly, for each of the involved referenced papers the main contribution that the authors of the referenced papers have brought to the current state of knowledge, the method used by the authors of the referenced papers, a brief presentation of the main obtained results and some limitations of the referenced article. This is the only way to contextualize the current state of the art in which the authors of the manuscript position their paper, identify and address aspects that have not been tackled/solved yet by the existing studies. The authors did not address this issue in the revised version of the manuscript even if I have signaled it in my previous review report.
4) The "Introduction" section. In the "Introduction" section, the authors must state more clearly a gap in the current state of knowledge that needs to be filled, a gap that is being addressed by their manuscript. At the end of the Introduction, the authors should present the structure of their paper, under the form: "The rest of the paper is structured as follows: Section 2 contains…". The authors did not address this issue in the revised version of the manuscript even if I have signaled it in my previous review report.
5) The "Materials and Methods" section. In the actual form of the manuscript, the "Materials and Methods" section is missing. It will benefit the paper if the authors restructure their paper and devise a proper "Materials and Methods" section, as requested by the Symmetry MDPI Journal's Template. I consider that the authors must pay more attention to the appropriate citation of the methods and results that have been retrieved from the scientific literature. When the authors present the information in the "Materials and Methods" section, they must assume clearly their original contribution by specifying this fact and by highlighting the fact that starting from a certain point there are depicted the original and novel aspects of their research. When presenting the devised method/approach in the "Materials and Methods" section, it will be extremely helpful to design and insert a diagram depicting the main sequence of steps that one has to process in order to reproduce the results of the conducted study. This diagram should be analyzed in detail within the manuscript by specifying all the elements needed for each and every step, in order to reach the final result of the study. The authors did not address this issue in the revised version of the manuscript even if I have signaled it in my previous review report.
6) The "Discussion" section. In order to validate the usefulness of their research, in the "Discussion" section that is currently missing from the manuscript, the authors should make a detailed comparison between their study from the manuscript and other ones that have been developed and used in the literature for the same or similar purposes. In the "Discussion" section the authors should also highlight current limitations of their study, and briefly mention some precise directions that they intend to follow in their future research work. The paper will benefit if the authors make a step further, beyond their approach and provide an insight at the end of the "Discussion" section regarding what they consider to be, based on the obtained results, the most important steps that all the involved parties should take in order to benefit from the results of the research conducted within the manuscript. The authors did not address this issue in the revised version of the manuscript even if I have signaled it in my previous review report.
Author Response
Title: New Fixed Point Theorems with Applications to Nonlinear Neutral Differential Equations
Authors: Jamshaid Ahmad, Laila A. Alnaser, Durdana Lateef and Hoda A. Fouad
Respected Referee
Please find attached an amended version of the manuscript with point-by point response to the reviewers’ comments. We are very grateful to you for your positive and helpful suggestions and we feel that the quality of the manuscript has been significantly improved as a result.
With Best Wishes
Dr. Jamshaid Ahmad
The authors have not provided a proper cover letter to my review report and a point-by-point response to my previous review report, while the majority of the signalled issues have not been addressed. In addition to these, the changes made within the manuscript have not been highlighted or track-changed.
I consider that the authors should improve their presentation, highlighting the originality and novelty of their approach in rapport with previous studies. The authors did not address this issue in the revised version of the manuscript even if I have signalled it in my previous review report.
I consider that the article will benefit if the authors take into account the following remarks and address within the manuscript the signalled issues:
1) The line numbers are missing in the whole pdf manuscript submitted by the authors, so I had to refer to excerpts of text and page numbers within the comments. The authors did not address this issue in the revised version of the manuscript even if I have signalled it in my previous review report.
Reply: We have tried to get the line numbers of each line but when we convert our paper into symmetry journal’s template, it disappears. We have tried more than 20 times for it. I hope you will understand the issue.
2) The sections of the manuscript. The manuscript under review will benefit if it is restructured in accordance with the Symmetry MDPI Journal's Template that provides a more logical structure that is much more appropriate for a research article. The restructuring of the manuscript will also help the authors to better express the novelty of their work and the contribution that they have made to the current state of knowledge. Consequently, the manuscript under review should be restructured as follows: Abstract, Keywords, 1. Introduction, 2. Materials and Methods, 3. Results, 4. Discussion, 5. Conclusions (not mandatory), 6. Patents (not mandatory), Supplementary Materials (not mandatory), Author Contributions, Funding, Acknowledgments, Conflicts of Interest, Appendices and References. Moreover, the authors must take into account the recommendations from the MDPI Symmetry Journal website regarding the format of the papers, by using the Microsoft Word template or LaTeX template to prepare their manuscript. In the actual form of the paper, most of these recommendations have not been taken into account and thus the reading of the manuscript is affected. The authors did not address this issue in the revised version of the manuscript even if I have signalled it in my previous review report.
Reply: We have added the missing sections and all the sections are now according to the format of the journal. Moreover, we have improved the presentation of the paper.
3) The "Introduction" section. I consider that the literature review should be improved by performing a careful analysis of the cited works. The authors must highlight exactly, for each of the involved referenced papers the main contribution that the authors of the referenced papers have brought to the current state of knowledge, the method used by the authors of the referenced papers, a brief presentation of the main obtained results and some limitations of the referenced article. This is the only way to contextualize the current state of the art in which the authors of the manuscript position their paper, identify and address aspects that have not been tackled/solved yet by the existing studies. The authors did not address this issue in the revised version of the manuscript even if I have signalled it in my previous review report.
Reply: We have performed careful analysis of the cited work. We have removed the unstated papers from the reference list. Now each reference describes a brief presentation of the obtained results in it.
4) The "Introduction" section. In the "Introduction" section, the authors must state more clearly a gap in the current state of knowledge that needs to be filled, a gap that is being addressed by their manuscript. At the end of the Introduction, the authors should present the structure of their paper, under the form: "The rest of the paper is structured as follows: Section 2 contains…". The authors did not address this issue in the revised version of the manuscript even if I have signalled it in my previous review report.
Reply: We have improved our paper and reduced the gaps in the current state of knowledge.
5) The "Materials and Methods" section. In the actual form of the manuscript, the "Materials and Methods" section is missing. It will benefit the paper if the authors restructure their paper and devise a proper "Materials and Methods" section, as requested by the Symmetry MDPI Journal's Template. I consider that the authors must pay more attention to the appropriate citation of the methods and results that have been retrieved from the scientific literature. When the authors present the information in the "Materials and Methods" section, they must assume clearly their original contribution by specifying this fact and by highlighting the fact that starting from a certain point there are depicted the original and novel aspects of their research. When presenting the devised method/approach in the "Materials and Methods" section, it will be extremely helpful to design and insert a diagram depicting the main sequence of steps that one has to process in order to reproduce the results of the conducted study. This diagram should be analyzed in detail within the manuscript by specifying all the elements needed for each and every step, in order to reach the final result of the study. The authors did not address this issue in the revised version of the manuscript even if I have signalled it in my previous review report.
Reply: We have added the “Materials and Methods” Section.
6) The "Discussion" section. In order to validate the usefulness of their research, in the "Discussion" section that is currently missing from the manuscript, the authors should make a detailed comparison between their study from the manuscript and other ones that have been developed and used in the literature for the same or similar purposes. In the "Discussion" section the authors should also highlight current limitations of their study, and briefly mention some precise directions that they intend to follow in their future research work. The paper will benefit if the authors make a step further, beyond their approach and provide an insight at the end of the "Discussion" section regarding what they consider to be, based on the obtained results, the most important steps that all the involved parties should take in order to benefit from the results of the research conducted within the manuscript. The authors did not address this issue in the revised version of the manuscript even if I have signalled it in my previous review report.
Reply: We have added the “Discussion” Section.
Round 3
Reviewer 2 Report
I have reviewed the second revised version of the manuscript "New Fixed Point Theorems with Applications to Nonlinear Neutral Differential Equations", Manuscript ID: symmetry-451449 and I can conclude that even if the authors did not manage to address all the signaled issues, overall the manuscript has been improved in contrast to the previous submissions.